# Prognostic Value of Severity Score Change for Septic Shock in the Emergency Room

**DOI:** 10.3390/diagnostics10100743

**Published:** 2020-09-24

**Authors:** Tae Sik Hwang, Hyun Woo Park, Ha Young Park, Young Sook Park

**Affiliations:** 1Department of Emergency Medicine, Inje University Haeundae Paik Hospital, Busan 48108, Korea; emhwang1@hanmail.net (T.S.H.); yourdoctor85@naver.com (H.W.P.); cprrosc@gmail.com (H.Y.P.); 2Department of Physical & Rehabilitation Medicine, Samsung Changwon Hospital, Sungkyunkwan University School of Medicine, Changwon 51353, Korea

**Keywords:** septic shock, prognosis, score, outcome, lactate

## Abstract

The vital signs or laboratory test results of sepsis patients may change before clinical deterioration. This study examined the differences in prognostic performance when systemic inflammatory response syndrome (SIRS), Sequential Organ Failure Assessment (SOFA), quick SOFA (qSOFA) scores, National Early Warning Score (NEWS), and lactate levels were repeatedly measured. Scores were obtained at arrival to triage, 1 h after fluid resuscitation, 1 h after vasopressor prescription, and before leaving the emergency room (ER) in 165 patients with septic shock. The relationships between score changes and in-hospital mortality, mechanical ventilation, admission to the intensive care unit, and mortality within seven days were compared using areas under receiver operating characteristic curve (AUROCs). Scores measured before leaving the ER had the highest AUROCs across all variables (SIRS score 0.827 [0.737–0.917], qSOFA score 0.754 [0.627–0.838], NEWS 0.888 [0.826–0.950], SOFA score 0.835 [0.766–0.904], and lactate 0.872 [0.805–0.939]). When combined, SIRS + lactate (0.882 [0.804–0.960]), qSOFA + lactate (0.872 [0.808–0.935]), NEWS + lactate (0.909 [0.855–0.963]), and SOFA + lactate (0.885 [0.832–0.939]) showed improved AUROCs. In patients with septic shock, scoring systems show better predictive performances at the timepoints reflecting changes in vital signs and laboratory test results than at the time of arrival, and combining them with lactate values increases their predictive powers.

## 1. Introduction

Sepsis accounts for about 30–50% of in-hospital deaths and often does not present with serious symptoms at the time of visit; therefore, 55.9% of cases involve normal blood pressure (BP) and lactate levels under 4 mmol/L [1]. Although the condition of sepsis patients may deteriorate rapidly or unexpected cardiac arrest may occur, early signs indicating the risk of deterioration do appear, which can be captured through routinely measured clinical data, such as vital signs or laboratory test results [2,3,4]. Scoring systems have been created for their own goal, and emergency room (ER) doctors use scoring systems to assess disease severity and predict prognosis in patients suspected of having infections or septicemia [5]. ER-based screening should focus not only on mortality but also on the need for an escalation of care so that mortality can be minimized [6].

Sepsis was previously identified and defined using the systemic inflammatory response syndrome (SIRS) criteria, though this definition was limited by its poor specificity [7,8]. SIRS was not developed as an early warning score but was designed to screen for and define sepsis, and SIRS criteria include parameters that are known to have limited predictive power for clinical deterioration [9]. The Third International Consensus Definition for Sepsis and Septic Shock (Sepsis 3) recommended the Quick Sequential Organ Failure Assessment (qSOFA) score to identify patients at high risk of death and prolonged intensive care unit (ICU) stay among those with suspected infection [10]. Meanwhile, the National Early Warning Score (NEWS) has been widely adopted in the United Kingdom as a tool to assess and monitor the clinical condition of hospitalized patients [11]. Given that the NEWS includes all variables from the qSOFA score and Between the Flag (BTF), patients identified to be at higher risk by these tools will have a similarly elevated NEWS. Therefore, aggregates of weighted scores, such as the NEWS, may be more accurate than single-parameter scores [12]. An increase in the SOFA score of 2 or more points within the first 24 h of ICU admission had superior prognostic accuracy for mortality and ICU stay for ≥3 days than the SIRS criteria or qSOFA score [13]. Quinten et al. evaluated whether repeated vital sign measurements in the ER can identify patients with sepsis or infection whose conditions will deteriorate within 72 h. They found that an increase in mean arterial pressure (MAP) over time was associated with a lower risk of deterioration, and a higher variability in MAP or an increase in respiratory rate over time, in combination with their respective values at ER admission, was associated with patient deterioration [14].

Various score systems were used to predict sepsis patients’ prognosis, mainly using the maximum value, or the initial or last value, over a certain period of time. This study was conducted to find out how comparison of scores before and after use of fluid resuscitation and vasopressor, which are key to early treatment of sepsis patients, affects the accuracy of prognosis predictions.

## 2. Materials and Methods

### 2.1. Patients

Medical records of adult patients aged 19 and older who visited the ER from January to June 2018 with confirmed sepsis based on bacterial culture (blood, urine, sputum, stool, or body fluid) and a mean arterial pressure (MAP) of less than 60 mmHg were analyzed. Blood culture sampling was performed before antibiotic prescription, and sputum, urine, stool, and body fluid cultures were all carried out within 24 h of antibiotic prescription.

This retrospective study was approved by the Inje University Haeundae Paik Hospital Institutional Review Boards (approved on 12 August 2020, 2020-07-040).

This emergency center is a secondary medical institution visited by 60,000 patients a year, with eight emergency medical specialists and four emergency medical residents working in shifts 24 h a day, and the treatment for septic shock patients is based on early goal-directed therapy. The goals were set as an MAP >65 mmHg, central venous pressure (CVP) of 8–12 cmH_2_O, urine output ≥0.5 mL/kg/h, and central venous oxygen saturation (ScvO_2_) of >70%; the items measured repeatedly at certain intervals were vital signs and laboratory test. If the MAP was <60 mmHg, two intravenous lines were secured and 30 mL/kg of fluid was rapidly injected; furthermore, an arterial line was secured and central venous catheterization and Foley catheterization were performed to monitor BP, CVP, and urine output. If the MAP had not recovered even after the initial fluid resuscitation, vasopressor injection was initiated with repeated fluid loading with 500 mL of crystalloid fluid. The time of vasopressor initiation depended on the decision of the emergency medical specialist. The amount of fluid loading in addition to the initial 30 mL/kg of crystalloid fluid was recorded.

### 2.2. Scores and Outcome Assessment

The NEWS, qSOFA scores, and SIRS scores were obtained four times: at the time of visiting the ER, during fluid resuscitation, at least 1 h after starting vasopressor administration, and before leaving the ER. The results of laboratory tests were replaced with the closest results when they did not match the exact time. The SOFA score was obtained twice: the first laboratory test results within 24 h and the laboratory test results after 24 h; in patients who left the ER within 24 h, the earliest results from the tests conducted after hospitalization were used. The primary outcome was mortality within seven days, and the secondary outcomes were in-hospital mortality, M/V, and ICU admission.

A study physician calculated the NEWS and qSOFA, SIRS, and SOFA scores using the vital signs documented in the electronic medical records. Additionally, leukocyte counts, partial pressure of CO_2_ (pCO_2_), and bandemia were used when available to calculate the SIRS score, and pO_2_, bilirubin levels, and creatinine levels were used when available to calculate the SOFA score. 

The qSOFA score is defined as a (1) systolic BP of ≤100 mmHg; (2) respiratory rate ≥22 breaths/min; and (3) altered mental status [10]. In this study, the Alert, Voice, Pain, and Unresponsive scale was used. Any state other than “Alert” was considered to indicate an altered mental status.

The NEWS is an aggregate score that has been validated as a risk indicator of clinical deterioration and mortality. It includes heart rate, systolic BP, respiratory rate, arterial oxygen saturation, any supplemental oxygen, temperature, and mental status, with each item weighing 0–3 points.

In the SIRS score, 1 point is assigned to each of the following items: body temperature >38 °C or <36 °C; pulse rate >90 bpm; respiratory rate >20 breaths/min or pCO_2_ < 32 mmHg; and leukocyte count >12,000/mm^3^ or <4000/mm^3^ or the presence of >10% immature neutrophils. The SIRS criteria are fulfilled with a score of ≥2 points.

The SOFA score includes the PaO_2_/FiO_2_ ratio derived from arterial blood gas analysis, platelet counts, Glasgow Coma Scale score, bilirubin levels, MAP, and creatinine levels, with each item weighing 0–4 points.

### 2.3. Statistical Analysis

Descriptive statistics were used to quantify baseline characteristics and patient outcomes. Data are presented as means ± standard deviations (SD) or medians with interquartile ranges (IQR) for continuous variables and frequencies and percentages for categorical variables. The accuracy in predicting outcome measures was assessed by calculating the area under the receiver operating characteristic curve (AUROC). The optimal cutoff point was chosen as the one that maximizes the Youden index (sensitivity + specificity-1). All statistical analyses were carried out using SPSS 21.0 software (IBM, Armonk, NY, USA) and R 3.4.4 (Vienna, Austria; http://www.R-project.org/). *p*-values less than 0.05 were considered statistically significant.

## 3. Results

### 3.1. Patient Characteristics

During the study period, 18,574 patients visited the ER: 332 cases of undifferentiated shock, 167 were excluded, and finally, 165 patients were included (Figure 1). The basic characteristics of the 165 patients are shown in Table 1. The mean age was 76 years, 57% were female (*n* = 87), and the median Charlson comorbidity index was 6. The details of initial treatment were as follows: the amount of additional fluid loading within 3 h was 2000 mL, the time to antibiotic prescription was 1.5 h, and the time to vasopressor administration was 3 h. The median SIRS and qSOFA score was 2, the median NEWS was 9, the median SOFA score was 8; and the median lactate level was 2.8 mmol/L.

### 3.2. Distribution of Patients According to Severity Scores

Figure 2 shows the distributions of severity scores of all 165 subjects according to the timing of measurement. The SIRS score, qSOFA score, NEWS, and lactate levels were all measured four times, and the SOFA score was measured only during triage at the ER and before leaving the ER. There was no significant difference in SIRS scores or lactate levels over time (*p* = 0.661 and *p* = 0.342, respectively). The qSOFA and SOFA scores significantly increased by 0.099 and 0.329 over time, respectively (*p* = 0.001 and *p* = 0.001, respectively). The NEWS significantly decreased by 0.928 over time (*p* < 0.001).

### 3.3. Sequential Changes in Severity Indicators

Table 2 shows the severity scores measured during triage at the ER, 1 h after fluid resuscitation, 1 h after vasopressor initiation, and before leaving the ER. The median SIRS and qSOFA values remained unchanged at 2 at all timepoints; the NEWS decreased from 9 to 6 points; the lactate levels decreased from 2.8 mmol/L to 2.0 mmol/L; and the SOFA scores increased from 8 to 9.

### 3.4. Predictive Values of Severity Scores and Lactate Levels for Mortality within Seven Days

Figure 3 shows the ROC curves and AUROCs for predicting mortality within seven days using the four severity scores and lactate levels at all timepoints. At triage at the ER, lactate levels had the highest AUROC of 0.771, followed by the SOFA score (0.711), qSOFA score (0.698), NEWS (0.680), and SIRS score (0.592). The cutoff values were 3 (SIRS), 2 (qSOFA), 10 (NEWS), 7 (SOFA), and 4 mmol/L (lactate). At the time of fluid resuscitation, the cutoff values of the SIRS score, qSOFA score, NEWS, and lactate level were 3, 2, 9, and 5 mmol/L, respectively, and the AUROCs of the SIRS score, qSOFA score, NEWS, and lactate level were 0.602, 0.686, 0.672, and 0.802, respectively, with lactate levels showing the highest value. At the time of vasopressor initiation, the cutoff values of the SIRS score, qSOFA score, NEWS, and lactate level were 2, 2, 7, and 3 mmol/L, respectively, and the corresponding AUROCs were 0.727, 0.690, 0.777, and 0.821, respectively. The cutoff values of the SIRS score, qSOFA score, NEWS, SOFA score, and lactate level before leaving the ER were 4, 2, 9, 11, and 3 mmol/L, respectively, and the corresponding AUROCs were 0.827, 0.754, 0.888, 0.835, and 0.827, respectively.

### 3.5. Predictive Values of Severity Scores and Lactate Levels for In-Hospital Mortality, Mechanical Ventilation, and ICU Admission

Figure 4 shows the ROC curves and AUROCs for in-hospital mortality, M/V, and ICU admission of all four severity scores and lactate levels. At triage at the ER, the SOFA score had the highest AUROC of 0.739, followed by lactate levels (0.732), NEWS (0.729), qSOFA score (0.704), and SIRS score (0.552). The cutoff values were 3 (SIRS), 2 (qSOFA), 7 (NEWS), 9 (SOFA), and 2 mmol/L (lactate). At the time of fluid resuscitation, lactate levels had the highest AUROC of 0.754, followed by the qSOFA score (0.727), NEWS (0.720), and SIRS score (0.540). The cutoff values were 3 (SIRS), 3 (qSOFA), 7 (NEWS), and 3 mmol/L (lactate). At the time of vasopressor initiation, the qSOFA score and NEWS had the highest AUROCs (both 0.822), followed by lactate levels (0.778) and the SIRS score (0.651). The cutoff values were 2 (SIRS), 3 (qSOFA), 7 (NEWS), and 2 mmol/L (lactate). Before leaving the ER, the SOFA score had the highest AUROC of 0.863, followed by the qSOFA score (0.831), NEWS (0.806), lactate levels (0.795), and SIRS (0.676). The cutoff values were 3 (SIRS), 3 (qSOFA), 5 (NEWS), 9 (SOFA), and 3 mmol/L (lactate).

### 3.6. Predictive Value of Maximum and Difference Value of Severity Score

There were no significant increases in the predictive value for mortality within seven days based on AUROCs when examining the maximum values, the differences between the values at ER triage and the maximum values, or the differences between the values at ER triage and the values on leaving the ER. Additionally, there were no increases in the predictive value, based on AUROCs, for in-hospital mortality, M/V, or ICU admission when examining the maximum values, the differences between the values at ER triage and the maximum values, or the differences between the values at ER triage and the values on leaving the ER (Table 3).

### 3.7. Predictive Value of Combination of Severity Score and Lactate

The combination of any severity score with lactate levels increased the AUROC at all four timepoints (two timepoints for SOFA) for mortality within seven days, and also increased the predictive value, based on AUROCs, for in-hospital mortality, M/V, or ICU admission (Table 4).

## 4. Discussion

The SIRS score, qSOFA score, NEWS, and SOFA score were measured at triage at the ER, 1 h after fluid resuscitation, 1 h after vasopressor initiation, and before leaving the ER in all patients with septic shock. The lactate level was the best predictor of mortality within seven days, and all four severity scores showed higher AUROCs before leaving the ER than at triage. As an exception, the AUROC of the NEWS was 0.888 before leaving the ER, which was higher than that of the lactate level (0.872). For all scores, the maximum value or the change in values between timepoints did not have a higher AUROC than that before leaving the ER. In other words, the maximum value or change in values between timepoints did not show an improved performance in predicting mortality within seven days compared to individual values. However, when lactate was combined with the severity scores, the AUROCs increased at all four timepoints, with the highest AUROC observed before leaving the ER for NEWS + lactate (0.909), followed by SOFA + lactate (0.885), SIRS + lactate (0.882), and qSOFA + lactate (0.872).

The highest AUROCs of severity scores for in-hospital mortality, M/V, and ICU admission were seen before leaving the ER, except for NEWS, which had the highest AUROC at the time of vasopressor initiation (0.822). The maximum AUROC for lactate levels was 0.800, which was higher than that at the time before leaving the ER (0.795). For all scores, there were no increases in the predictive value, based on AUROCs, for in-hospital mortality, M/V, or ICU admission when examining the maximum values, the differences between the values at ER triage and the maximum values, or the differences between the values at ER triage and the values on leaving the ER compared to individual values.

When lactate levels were combined with the severity scores, the AUROCs increased at all four timepoints, and the highest AUROC at the time before leaving the ER was observed for qSOFA + lactate, followed by SOFA + lactate, NEWS + lactate, and SIRS + lactate.

The target patients were prescribed both antibiotics and vasopressors, including fluid resuscitation. Therefore, rather than determining disposition within a short period of time, it is necessary to measure vital signs repeatedly over the course of treatment, observe the responses to treatment, and repeat laboratory tests. For predicting a poor prognosis, none of the scoring systems were particularly excellent, but performance was improved in all scoring systems on combination with lactate levels. As an ER tends to be overcrowded and lacking in personnel, triage inevitably proceeds with simple tools and preferably, a relatively easy method for the escalation of treatment or determination of disposition.

ER clinicians use sepsis screening tools for broader purposes, such as for identifying critically ill patients, initiating critical and time-sensitive interventions, and determining the need for ICU care. Identifying patients with critical illness is paramount in the emergency care of patients with infectious diseases. Highly sensitive prediction tools will allow clinicians to identify nearly all critically ill patients early in their treatment course and will enable the provision of time-sensitive interventions such as fluid resuscitation, early antibiotic administration, and timely vasopressor administration. Highly specific predictive tools allow for more judicious disposition decisions and expenditure of resources, particularly ICU beds [15]. Therefore, to improve the performance of a scoring system, a combination is recommended rather than using the scoring system alone. Jo et al. reported that NEWS + lactate can provide excellent discriminating value in predicting two-day mortality in general ER patients, and it has the best discriminating value regarding the need for critical care and composite outcomes [16]. Baumann et al. reported that the combination of qSOFA criteria with initial lactate levels provides substantially improved sensitivity for the screening of critical illness compared to isolated lactate and qSOFA thresholds [6]. This study also showed a marked improvement in performance when lactate was combined with each of the four scoring systems.

While using a single scoring system or using a system in combination with laboratory test results, the worst value is usually used during ER stay; however, studies have shown that repeated measurements improve performance. Quinten et al. evaluated whether repeated vital sign measurements in the ER can be used to identify patients with sepsis or infection whose conditions will deteriorate within 72 h; the inclusion of repeated MAP measurements resulted in the highest AUC (0.800) [14]. Schulte-Hubbert et al. reported that the BP criterion of the CRB-65 score (used in assessing pneumonia severity) is rarely met at hospital admission, but within the next 24 h, a drop in BP below the score threshold occurs in nearly half of the patients. Such a decrease is associated with future clinical deterioration and the need of M/V or vasopressin with a high sensitivity [17]. One study demonstrated that prehospital hemodynamic variability was associated with clinical deterioration [18]. In another study, repeated BP measurements in the ER were related to improved outcome prediction compared to single measurements, but in the same study, repeated measurements of other vital signs failed to improve the prognostic ability [14]. It remains unknown whether changes in the Modified Early Warning Score (MEWS) in the ER are correlated with outcomes or improve outcome predictions compared to single or static measures.

Levin et al. reported that dynamic vital signs categorized by the MEWS in the ER, specifically abnormalities that failed to normalize, were associated with increased mortality, a higher probability of ICU admission, longer length of hospital stay, and sepsis [19]. Changes in the MEWS during ER stay were superior to static scores at triage in predicting mortality, ICU admission, and sepsis. The final MEWS score in the ER was strongly associated with outcomes and was comparable to delta MEWS scores for the prediction of death and ICU admission; this suggested that the condition in which patients leave the ER is as important as the changes in their condition in the ER. In this study, the predictive performance of each scoring system differed over time. Measurements taken before leaving the ER were the best; thus, in patients with septic shock, the determination of disposition using severity scores obtained after fluid resuscitation and antibiotic and vasopressor prescription is expected to be helpful.

The limitations of this study are as follows. First, it was performed at a single medical institution with a small number of patients, and thus, it is not fully representative of all patients. Second, there was a potential for selection bias as this was a retrospective study. Third, although a protocol was created and implemented for shock management, the only parameter among the set goals that was repeatedly measured was MAP. As the studied patients had septic shock, there were no missing data on vital signs or lactate levels, except in two cases.

## 5. Conclusions

NEWS plus lactate was the best for predicting mortality within seven days in sepsis patients admitted to the emergency room, and qSOFA plus lactate and SOFA plus lactate were good for predicting in-hospital mortality, application of mechanical ventilation, and admission to intensive care unit. As for the point of score measurement, the performance of the values measured at the time of leaving the emergency room after all of the initial treatments for sepsis, such as fluid resuscitation, vasopressor, and antibiotics, were performed.

## Figures and Tables

**Figure 1 diagnostics-10-00743-f001:**
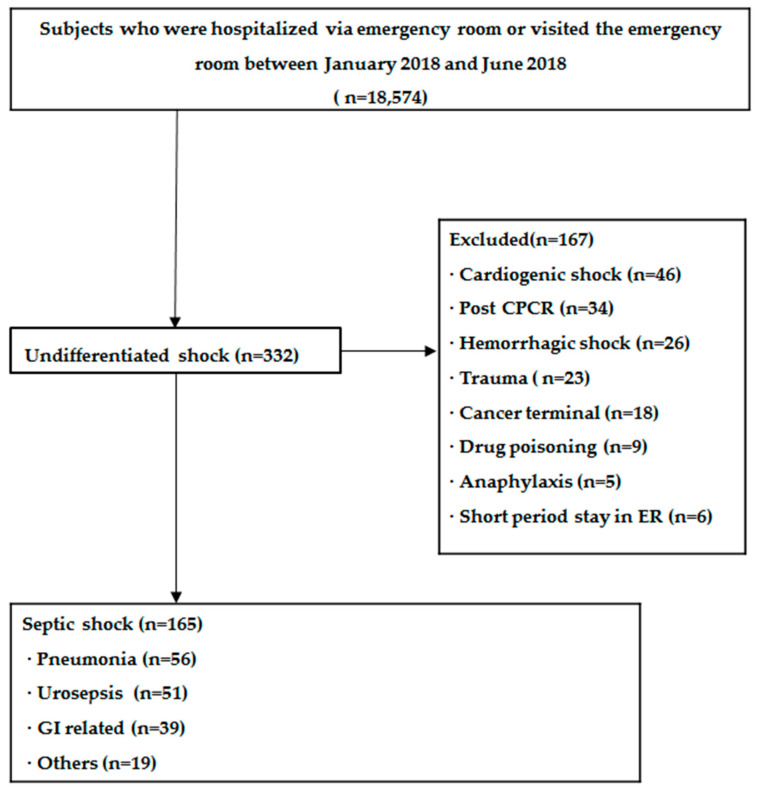
Flow chart of subjects.

**Figure 2 diagnostics-10-00743-f002:**
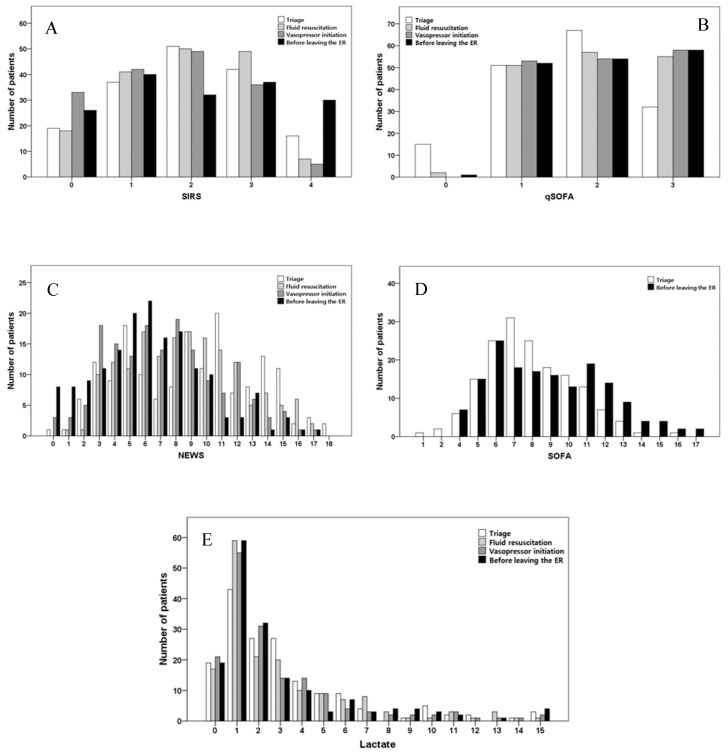
Distribution of patients according to severity score and lactate level. (**A**) Systemic inflammatory response syndrome (SIRS). (**B**) Quick Sequential Organ Failure Assessment (qSOFA). (**C**) National Early Warning Score (NEWS). (**D**) Sequential Organ Failure Assessment (SOFA). (**E**) Lactate.

**Figure 3 diagnostics-10-00743-f003:**
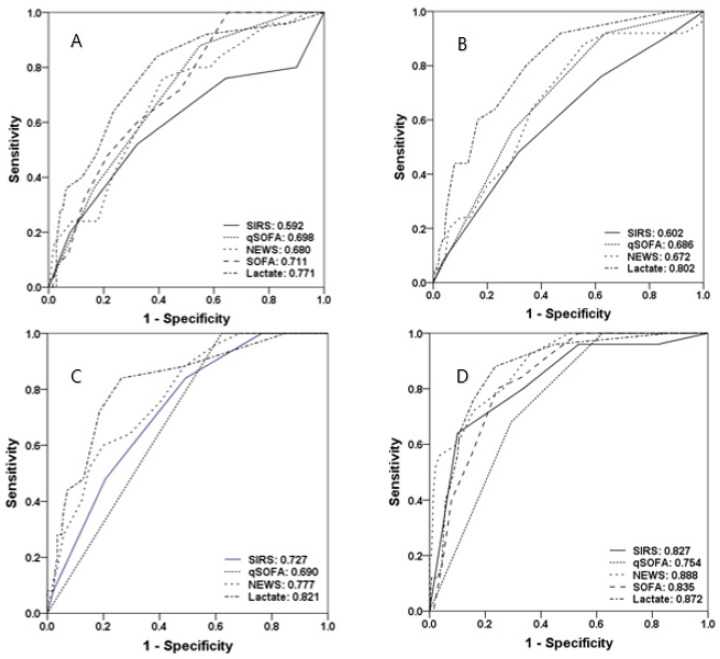
Areas under receiver operating characteristic curves for the prediction of mortality within seven days of the SIRS score, qSOFA score, NEWS, SOFA score, and lactate level across four timepoints (triage (**A**), fluid resuscitation (**B**), initiation of vasopressor (**C**), and before leaving ER (**D**)). SIRS, systemic inflammatory response syndrome; SOFA, Sequential Organ Failure Assessment; qSOFA, quick SOFA; NEWS, National Early Warning Score.

**Figure 4 diagnostics-10-00743-f004:**
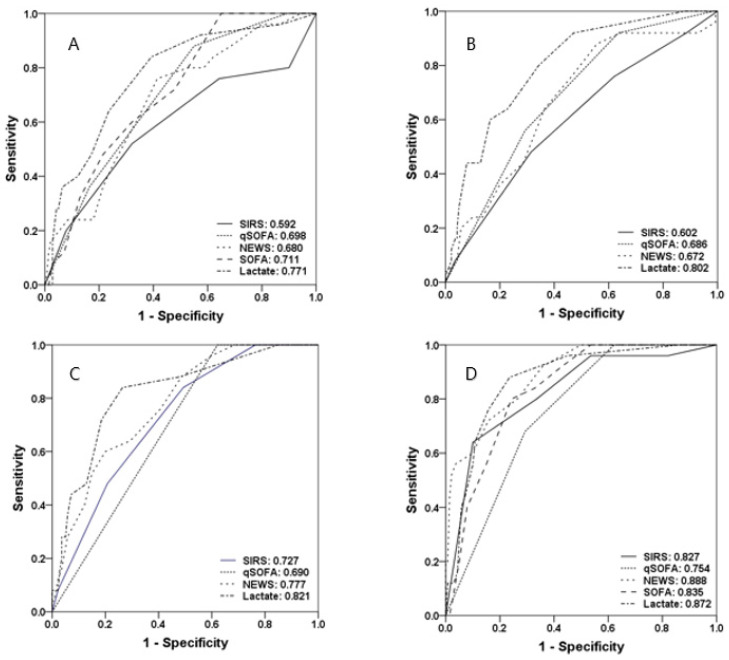
Areas under receiver operating characteristic curves for the prediction of in-hospital mortality, application of mechanical ventilation, and admission to the ICU of the SIRS score, qSOFA score, NEWS, SOFA score, and lactate levels across four timepoints (triage (**A**), fluid resuscitation (**B**), start vasopressor (**C**), and before leaving the ER (**D**)). ICU, intensive care unit; SIRS, systemic inflammatory response syndrome; SOFA, Sequential Organ Failure Assessment; qSOFA, quick SOFA; NEWS, National Early Warning Score.

**Table 1 diagnostics-10-00743-t001:** Baseline characteristics of the patients.

Characteristic	Value
Sex (%)	
Female	87 (53)
Male	78 (47)
Age (year), median (IQR)	76 (64–84)
Charlson comorbidity index	6 (5–7)
Final diagnosis (%)	
Pneumonia	56 (34)
Urosepsis	51 (31)
Intra-abdominal infection	39 (24)
Others	19 (11)
Procalcitonin (ng/mL)	4.24 (0.99–24.13)
Duration of symptoms, median (IQR)(hour)	24 (7–48)
Initial management	
Fluid loading (mL)	2000 (1500–2500)
Time to antibiotic prescription, median (IQR)(hour)	1.5 (0.5–2.5)
Time to the initiation of vasopressors (norepinephrine), median (IQR)(hour)	3 (2–6)
Disposition	
Length of ER stay (hours)	21 (10–34)
Length of hospital stay (days)	13 (7–20)
Outcome (%)	
Mortality within 7 days	25 (15)
In-hospital mortality	45 (27)
Use of M/V	48 (29)
Admission to the ICU	74 (45)
SIRS score ≥2 (%)	109 (66)
SIRS score, median (IQR)	2 (1–3)
qSOFA score ≥2 (%)	99 (60)
qSOFA score, median (IQR)	2 (1–2)
NEWS ≥5 (%)	136 (82)
NEWS, median (IQR)	9 (5–12)
SOFA score ≥2 (%)	164 (99)
SOFA score, median (IQR)	8 (6–10)
Lactate ≥4 mmol/L (%)	103 (62)
Lactate, median (IQR)	2.8 (1.4–4.4)

IQR, interquartile range; ER, emergency room; M/V, mechanical ventilation; ICU, intensive care unit; SIRS, systemic inflammatory response syndrome; SOFA, Sequential Organ Failure Assessment; qSOFA, quick SOFA; NEWS, National Early Warning Score.

**Table 2 diagnostics-10-00743-t002:** Sequential changes in severity indicators.

	Time (Hour)	SIRS	qSOFA	NEWS	SOFA	Lactate (mmol/L)
Triage						
Mean ± SD	-	1.99 ± 1.16	1.70 ± 0.88	8.96 ± 4.25	7.89 ± 2.46	3.59 ± 3.14
Median (IQR)	-	2 (1–3)	2 (1–2)	9 (5–12)	8 (6–10)	2.8 (1.4–4.4)
Fluid resuscitation						
Mean ± SD	2.32 ± 1.75	1.92 ± 1.07	2.00 ± 0.83	8.70 ± 3.66	-	3.41 ± 3.15
Median (IQR)	2 (1–3)	2 (1–3)	2 (1–3)	9 (6–11)	-	2.4 (1.2–4.4)
Initiation of vasopressors						
Mean ± SD	5.89 ± 3.34	1.62 ± 1.12	2.03 ± 0.82	7.27 ± 3.69	-	3.15 ± 3.06
Median (IQR)	5 (4–7)	2 (1–2)	2 (1–3)	7 (4–10)	-	2.0 (1.1–4.0)
Before leaving the ER						
Mean ± SD	11.11 ± 4.72	2.03 ± 1.35	2.02 ± 0.83	6.34 ± 3.65	8.88 ± 3.08	3.24 ± 3.25
Median (IQR)	10 (8–14)	2 (1–3)	2 (1–3)	6 (4–8)	9 (6–11)	2.0 (1.0–3.9)
Maximum	-	4	3	18	17	15
Δ from the triage value to the maximum value	-	0.75 ± 0.90	0.56 ± 0.65	1.55 ± 2.22	1.37 ± 1.75	0.82 ± 1.63
Δ from the triage value to the value measured before leaving the ER	-	0.04 ± 1.49	0.32 ± 0.76	−2.62 ± 3.88	0.99 ± 2.19	−0.36 ± 2.35

SD, standard deviation; IQR, interquartile range; ER, emergency room; M/V, mechanical ventilation; ICU, intensive care unit; SIRS, systemic inflammatory response syndrome; SOFA, Sequential Organ Failure Assessment; qSOFA, quick SOFA; NEWS, National Early Warning Score; **Δ,** difference.

**Table 3 diagnostics-10-00743-t003:** Comparisons of areas under receiver operating characteristic curves with 95% confidence intervals for the prediction of mortality within seven days, in-hospital mortality, M/V, admission to ICU by maximum and difference value of severity score.

	Maximum	From the Triage Value to the Maximum Value (Δ)	From the Triage Value to the Value Measured before Leaving the ER (Δ)
AUROCs for the prediction of mortality within 7 days			
SIRS	0.768 (0.661–0.876)	0.680 (0.564–0.797)	0.746 (0.652–0.839)
qSOFA	0.718 (0.627–0.809)	0.513 (0.394–0.632)	0.551 (0.435–0.666)
NEWS	0.745 (0.652–0.838)	0.576 (0.452–0.700)	0.707 (0.592–0.821)
SOFA	0.811 (0.735–0.887)	0.746 (0.649–0.843)	0.764 (0.679–0.848)
Lactate	0.848 (0.777–0.919)	0.723 (0.603–0.843)	0.622 (0.472–0.772)
AUROCs for the prediction of in-hospital mortality, M/V, admission to ICU			
SIRS	0.641 (0.556–0.727)	0.592 (0.506–0.679)	0.634 (0.550–0.719)
qSOFA	0.773 (0.699–0.847)	0.535 (0.446–0.624)	0.602 (0.516–0.688)
NEWS	0.795 (0.727–0.863)	0.534 (0.445–0.622)	0.521 (0.433–0.609)
SOFA	0.843 (0.783–0.903)	0.725 (0.647–0.803)	0.741 (0.665–0.817)
Lactate	0.800 (0.733–0.867)	0.678 (0.597–0.758)	0.544 (0.457–0.632)

ER, emergency room; SIRS, systemic inflammatory response syndrome; SOFA, Sequential Organ Failure Assessment; qSOFA, quick SOFA; NEWS, National Early Warning Score.

**Table 4 diagnostics-10-00743-t004:** Comparisons of areas under receiver operating characteristic curves with 95% confidence intervals for the prediction of mortality within seven days, in-hospital mortality, M/V, admission to ICU by combination of severity score and lactate.

	ER Triage	Fluid Resuscitation	Initiation of Vasopressor	Before Leaving ER
AUROCs for the prediction of mortality within 7 days				
SIRS + lactate	0.764 (0.661–0.867)	0.798 (0.710–0.886)	0.825 (0.742–0.908)	0.882 (0.804–0.960)
qSOFA + lactate	0.787 (0.697–0.877)	0.827 (0.758–0.897)	0.830 (0.758–0.903)	0.872 (0.808–0.935)
NEWS + lactate	0.757 (0.655–0.859)	0.805 (0.721–0.890)	0.844 (0.768–0.920)	0.909 (0.855–0.963)
SOFA + lactate	0.809 (0.728–0.891)			0.885 (0.832–0.939)
AUROCs for the prediction of in-hospital mortality, M/V, admission to ICU				
SIRS + lactate	0.728 (0.651–0.805)	0.738 (0.662–0.814)	0.799 (0.731–0.867)	0.815 (0.751–0.880)
qSOFA + lactate	0.776 (0.707–0.845)	0.822 (0.760–0.885)	0.871 (0.818–0.924)	0.882 (0.831–0.933)
NEWS + lactate	0.768 (0.697–0.838)	0.801 (0.736–0.867)	0.854 (0.797–0.910)	0.854 (0.799–0.910)
SOFA + lactate	0.798 (0.731–0.864)			0.881 (0.829–0.932)

ER, emergency room; SIRS, systemic inflammatory response syndrome; SOFA, Sequential Organ Failure Assessment; qSOFA, quick SOFA; NEWS, National Early Warning Score.

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
