# Peer review of "Prognostic Value of Severity Score Change for Septic Shock in the Emergency Room"

_diagnostics, 2020, doi:10.3390/diagnostics10100743_

Round 1
Reviewer 1 Report
This is an interesting paper examining scores which may be helful to predict the outcome of spetic patients presenting to the ER - albeit in a retrospective study, in a small group of patients in a single centre.
Minor comments:
1) the paper would benefit from English sub-editing.
2) the introduction contains some elements of the results - lines 61-68 - which should be moved the appropriate section.
3) the paper would benefir from being shortened by 30% by removing all repetition.
4) the results section has major repetition between the text, figures and tables. Data should be represented only once in one of the formats (text OR figure OR table) - not in all three.
5) in the conclusion it would be helpful if the authors could make a recommendation about which score (plus lactate) and at which time-point to do it to get the best predictive value in this group of patients in the ER.
Author Response
Comments and Suggestions for Authors
Thank you for your review and kind comments and for giving us the opportunity to revise them.
This is an interesting paper examining scores which may be helful to predict the outcome of spetic patients presenting to the ER - albeit in a retrospective study, in a small group of patients in a single centre.
Minor comments:
1) the paper would benefit from English sub-editing.
: This research paper was edited by a professional English editing company called editage.
2) the introduction contains some elements of the results - lines 61-68 - which should be moved the appropriate section.
: This study aimed to assess whether there were differences in the prognostic performances of the SIRS, qSOFA, NEWS, and SOFA scores, which are frequently used for severity evaluation, when repeated measurements of these scores were carried out. For sepsis patients who received both fluid resuscitation and vasopressor prescriptions, the SIRS, qSOFA, NEWS, and SOFA scores were repeatedly obtained based on vital signs and laboratory test results measured during ER admission. The relationships between individual values, changes in these values, in-hospital mortality within 7 days, ICU admission, mechanical ventilation (M/V), and length of stay were evaluated. It was found that scoring systems have better predictive performances at the time points reflecting changes in vital signs and laboratory test results than at the time of arrival, and combining these scores with lactate values increases their predictive powers.
-> Various score systems were used to predict sepsis patients' prognosis, mainly using the maximum value, or the initial or last value, over a certain period of time. This study was conducted to find out how comparison of scores before and after use of fluid resuscitation and vasopressor, which are key to early treatment of sepsis patients, affects the accuracy of prognosis predictions.
3) the paper would benefir from being shortened by 30% by removing all repetition.
4) the results section has major repetition between the text, figures and tables. Data should be represented only once in one of the formats (text OR figure OR table) - not in all three.
-> As you pointed out, we eliminated repetitive content.
5) in the conclusion it would be helpful if the authors could make a recommendation about which score (plus lactate) and at which time-point to do it to get the best predictive value in this group of patients in the ER.
-> For predicting the prognosis of patients with septic shock, scoring systems show better performances at time points reflecting changes in vital signs and laboratory test results due to treatment than at the time of arrival at the ER. The combination of lactate with these scoring systems increases their predictive powers.
NEWS plus lactate was the best for predicting mortality within 7 days in sepsis patients admitted to the emergency room, and qSOFA plus lactate and SOFA plus lactate were good for predicting in-hospital mortality, application of mechanical ventilation, and admission to intensive care unit. As for the point of score measurement, the performance of the values measured at the time of leaving the emergency room after all of the initial treatments for sepsis, such as fluid resuscitation, vasopressor, and antibiotics, were performed.
Reviewer 2 Report
The authors used the combine scoring system with lactate laboratory testing as predictors for clinical status of sepsis patients. Presentation of data need to be more comprehensive. Need to give separate graphical representation of lactate level to compare it with other score system. In table 1 add percentage sign (%) to percentage values to make data easily understandable.
Author Response
Comments and Suggestions for Authors
Thank you for your review and kind comments and for giving us the opportunity to revise them.
The authors used the combine scoring system with lactate laboratory testing as predictors for clinical status of sepsis patients. Presentation of data need to be more comprehensive. Need to give separate graphical representation of lactate level to compare it with other score system. In table 1 add percentage sign (%) to percentage values to make data easily understandable.
- As pointed out, Table 3 and Table 4 were separated to highlight the lactate-related results, and the percentage sign (%) and percentage values in Table 1 were corrected in red.